# Quantifying the polygenic contribution to variable expressivity in eleven rare genetic disorders

M.T. Oetjens[1]*, M.A. Kelly[1], A.C. Sturm[1], C.L. Martin[1] & D.H. Ledbetter [1]*

Rare genetic disorders (RGDs) often exhibit significant clinical variability among affected individuals, a disease characteristic termed variable expressivity. Recently, the aggregate effect of common variation, quantified as polygenic scores (PGSs), has emerged as an effective tool for predictions of disease risk and trait variation in the general population. Here, we measure the effect of PGSs on 11 RGDs including four sex-chromosome aneuploidies (47, XXX; 47,XXY; 47,XYY; 45,X) that affect height; two copy-number variant (CNV) disorders (16p11.2 deletions and duplications) and a Mendelian disease (melanocortin 4 receptor deficiency (*MC4R*)) that affect BMI; and two Mendelian diseases affecting cholesterol: familial hypercholesterolemia (FH; *LDLR* and *APOB*) and familial hypobetalipoproteinemia (FHBL; *PCSK9* and *APOB*). Our results demonstrate that common, polygenic factors of relevant complex traits frequently contribute to variable expressivity of RGDs and that PGSs may be a useful metric for predicting clinical severity in affected individuals and for risk stratification.

[1] Geisinger Health System, Danville, PA, USA. *email: mtoetjens@geisinger.edu; dhledbetter@geisinger.edu

Most rare genetic disorders (RGDs) exhibit some degree of variability in phenotypic presentation among affected individuals[1–4]. Phenotypes of individuals with the same rare, pathogenic variant can range from mild features that may never cause clinical concern, to those with severe clinical manifestations that impact quality of life and reduce life expectancy. Often, the modifiers of pathogenic variants are assumed to be other genetic and/or unspecified environmental factors; however, they are rarely identified. Many of the primary symptoms of RGDs are actually extremes of normally-distributed phenotypes in the general population. In traits with both monogenic and polygenic inheritance, there is emerging evidence that the common alleles that explain variance of the trait in the general population may also contribute to variable expressivity in some RGDs[5].

Previous studies of RGDs have examined the contribution of familial background on variable expressivity. Turner syndrome (45,X) is an RGD with short stature as a phenotypic hallmark caused by loss of the pseudoautosomal short-stature homeobox-containing gene (SHOX), which is essential for normal skeletal maturation and growth plate fusion development[6]. Even though height is severely affected in 45,X, parental height still correlates with affected probands at similar coefficients as their unaffected siblings[7]. The importance of familial background on variable expressivity has also been demonstrated in males with increased height caused by the presence of an extra X-chromosome (47, XXY)[7]. Since height is primarily determined by genetic contributions from common variants, this observation suggests that at least a component of the heritability can remain intact even when one copy of a haploinsufficient or triplosensitive locus in the core biological pathway is lost or gained, respectively[8]. Furthermore, our previous study of individuals with a 16p11.2 de novo deletion, which increases the risk for cognitive, motor, and behavior deficits, demonstrated that parental phenotypes (biparental mean) are predictive of the proband's phenotype across these dimensions[9]. We also observed the expected large effect of a de novo 16p11.2 deletion on body-mass index (BMI)[10] and showed a trend between proband BMI and the biparental mean[9].

Polygenic scores (PGSs) measure the cumulative effect of common alleles identified in genome-wide association studies (GWAS) and allow for a direct assessment of an individual's genetic predisposition for phenotypic expression or risk of disease[11,12]. For coronary artery disease (CAD) and breast cancer, PGSs have been reported to provide a substantial level of risk stratification in the general population and have generated a considerable amount of debate about their potential use as a clinical risk prediction tool[13,14]. This debate stems in part from reports of an equivalence in disease risk between an individual having an extreme PGS or a rare pathogenic variant. However, few studies have systematically evaluated extreme PGSs and rare pathogenic variants in the same study population for a comparison of their relative phenotypic impacts. If equivalent, these data could provide further justification for the clinical utility of PGSs in circumstances where rare variant testing is already incorporated into clinical care.

Recently, to explore variable expressivity, several studies have examined whether PGSs can explain trait variance and disease prevalence in individuals who are already at substantially elevated risk due to an RGD. For example, a study of clinically ascertained patients with autosomal dominant familial hypercholesterolemia (FH) caused by pathogenic variants in the low-density lipoprotein receptor gene (LDLR) revealed that a LDL-C PGS ($PGS_{LDL-C}$) constructed of common, genome-wide significant markers modified the low-density lipoprotein cholesterol (LDL-C) phenotype[15,16]. Considering the critical role LDLR plays in the regulation of plasma LDL-C, this finding confirms that common

variation can contribute to variable expressivity of severe RGDs. Similar results have been demonstrated in schizophrenia, familial breast cancer, neurodevelopmental disorders, and Alzheimer's disease[17–20]. Despite these promising examples, for most RGDs there are limited data on the applicability of PGSs for predictions of clinical severity, despite a pressing need.

Here, we use a genotype-first approach and leverage the MyCode™ Community Health Initiative cohort ($n = 92,455$), a health system-based population, to identify and measure the effect size of pathogenic rare variants underlying 11 RGDs identified through the Geisinger-Regeneron DiscovEHR Collaboration, which generates exome sequence data from the MyCode cohort[21]. Although RGDs are formally defined as affecting fewer than 1 in 2000 individuals in the US[22], exceptions have been made for some highly penetrant variants (e.g. Trisomy 21, 1 in 600 individuals[23]); therefore, for this study we include some pathogenic rare variants with a frequency greater than 1 per 2000 individuals in our definition of RGDs. We select 11 RGDs that affect three highly heritable and routinely measured quantitative traits: height (sex-chromosome aneuploidies including 45,X; 47, XXX; 47,XXY; and 47,XYY), BMI (MC4R and 16p11.2 deletions and duplications)[10,24]. These traits are clinically relevant in well-characterized and relatively prevalent RGDs and understanding their variable expressivity may impact clinical decision making. The focus of the present study is to explain variable expressivity within a single genetic etiology. To that end, we limit our analysis to RGDs prevalent enough in the MyCode cohort that we have at least 50% power to detect a nominal association in a linear model. We develop three PGSs from the summary statistics of external GWAS data ($PGS_{LDL-C}$, $PGS_{HEIGHT}$, and $PGS_{BMI}$) and optimize them using our validation cohort of 10,000 variant-negative patients from our health system-based population[25,26]. With optimized PGSs, we compare the effect size of an extreme PGS to RGD-causing variants to provide insight into their relative clinical utility. Lastly, using a working definition of variable expressivity as trait-variance among individuals with functionally equivalent RGD-causing variants, we quantify the polygenic contribution with linear models using PGS as predictors across the 11 RGDs.

## Results

**Effect sizes of RGD-causing variants.** We identified 609 unrelated individuals in DiscovEHR with rare pathogenic variants underlying one of the 11 RGDs and meeting our sample inclusion criteria. In the testing cohort (average age at last visit = 60.42 years; 57.7% female), the mean (SD) of the three quantitative phenotypes were LDL-C: 137.67 mg dl^-1 (41.63); female height 161.94 cm (6.66); male height: 176.69 cm (7.20); and BMI 31.94 kg m^−2 (7.72). The phenotypes of patients with RGD-causing variants were significantly different than variant-negative patients ($n = 31,430$; Supplementary Fig. 1), in accordance with the known clinical presentation of the RGDs (Table 1).

First, we tested for associations between sex-adjusted height and four sex-chromosome aneuploidies. Our study included 42 females with 47,XXX; 44 males with 47,XXY; 24 males with 47, XYY; and 19 females with 45,X or 45,X/46,XX sex-chromosome complements. Individuals with an extra X-chromosome, 47,XXX and 47,XXY, were 0.93 SD (95% Confidence Interval [CI]: 0.64, 1.23) and 0.56 SD (95% CI: 0.26, 0.85) taller than euploid individuals, respectively. The 47,XYY complement had the largest increase in height among sex-chromosome aneuploidies as these patients were 1.32 SD (95% CI: 0.92,1.72) taller than euploid individuals. Of the 19 patients with 45,X or 45,X/46,XX, the median logR Ratio (mLRR) and B-allele frequency (BAF) profiles of six patients were consistent with complete to nearly complete loss of the X-chromosome, as defined by > 80% of cells having 45,

**Table 1 Effect sizes of rare pathogenic variants underlying rare genetic disorders**

| Trait | RGD | Beta (95% CI) | P-Value | Sample Size |
|---|---|---|---|---|
| Height | 47,XXX | 0.93 (0.64, 1.23) | $1.46 \times 10^{-9}$ | 42 |
| | 47,XXY | 0.56 (0.26, 0.85) | $2.22 \times 10^{-4}$ | 44 |
| | 47,XYY | 1.32 (0.92, 1.72) | $8.67 \times 10^{-11}$ | 24 |
| | 45,X | −1.91 (−2.37, −1.47) | $6.52 \times 10^{-17}$ | 19 |
| BMI | *MC4R* Deficiency | 0.64 (0.39, 0.90) | $9.03 \times 10^{-7}$ | 58 |
| | 16p11.2 Deletion | 1.34 (1.05, 1.64) | $4.80 \times 10^{-19}$ | 44 |
| | 16p11.2 Duplication | −0.52 (−0.80, −0.25) | $2.09 \times 10^{-4}$ | 50 |
| LDL-C | *LDLR* FH | 2.49 (2.33, 2.65) | $1.15 \times 10^{-208}$ | 146 |
| | *APOB* FH | 1.42 (1.21, 1.62) | $7.39 \times 10^{-42}$ | 87 |
| | *PCSK9* FHBL | −0.72 (−1.01, −0.43) | $1.55 \times 10^{-6}$ | 42 |
| | *APOB* FHBL | −1.59 (−1.86, −1.33) | $8.49 \times 10^{-33}$ | 53 |

*RGD* Rare Genetic Disorder
*CI* Confidence Interval
*FH* Familial Hypercholesterolemia
*FHBL* Familial Hypobetalipoproteinemia

X (Supplementary Fig. 2 and Supplementary Fig. 3). The remaining 13 patients were mosaic and the proportion of 45,X in these individuals fell between 60 and 80% of the cells. Individuals with 45,X and 45,X/46,XX were shorter than euploid individuals by −2.54 and −1.63 SD, respectively. For sample size considerations, 45,X and 45,X/46,XX were analyzed together and as a group were −1.91 SD (95% CI: −2.37, −1.47) shorter than euploid individuals. For simplicity, the combined 45,X and 45,X/46,XX sample is referred to as 45,X in the figures and the remainder of the text unless otherwise noted.

Next, we identified 58 individuals with rare pathogenic *MC4R* variants, 44 16p11.2 deletions, and 50 16p11.2 duplications that cause familial or de novo forms of obesity or leanness. The presence of a rare pathogenic *MC4R* variant was associated with a 0.64 (95% CI: 0.39, 0.90) SD increase in BMI. We also observed the widely reported dosage effect of the 16p11.2 copy number variation (CNV) on BMI, where the deletion and duplication predispose affected individuals to obesity and leanness, respectively[10,27]. In DiscovEHR, the BMIs of patients with 16p11.2 deletions were 1.34 SD (95% CI: 1.05, 1.64) higher and 16p11.2 duplications were −0.52 SD (95% CI: −0.80, −0.25) lower than variant-negative individuals from the general population.

Lastly, we identified 146 individuals with a pathogenic *LDLR* variant and 87 individuals with a pathogenic *APOB* variant. Consistent with the severe phenotype of FH, the maximum documented LDL-C in individuals with pathogenic *LDLR* or *APOB* variants was 2.49 SD (95% CI: 2.33, 2.65) and 1.42 SD (95% CI: 1.21, 1.62) higher than variant-negative individuals from the general population, respectively. We also identified 95 patients with FHBL-causing variants, pLOFs in *PCSK9* or *APOB*, which were associated with a decrease of LDL-C by −0.72 SD (95% CI: −1.01, −0.43) and −1.59 SD (95% CI: −1.86, −1.33), respectively.

**Variance explained by PGSs in DiscovEHR.** To optimize PGSs for the testing cohort, we identified the ρ tuning parameters that maximized the variance explained by LDPred in the validation cohort for each quantitative trait (Supplementary Table 1). The performance of the optimized PGSs were consistent across validation and testing cohorts and were strong predictors of the respective quantitative traits (Table 2). In the testing cohort, the variance explained by $PGS_{HEIGHT}$ (21.71%) and $PGS_{BMI}$ (10.78%) were similar to those reported in the combined GWAS meta-analysis publication that produced the summary statistics (Supplementary Notes 1 and 2)[26]. The variance explained by our $PGS_{LDL-C}$ in the testing cohort was 7.99%.

We binned individuals into 100 groups based on percentiles of the $PGS_{HEIGHT}$ to examine phenotypes at the median and tails of the distribution (Fig. 1a). The average height of 46,XY males and 46,XX females in the 50th percentile bin was 177.45 cm and 162.04 cm, respectively, which is consistent with the U.S. national average reported by the Center for Disease Control[28]. The difference in the mean height between the 1st (bottom 1%) and 100th (top 1%) percentile bins of the $PGS_{HEIGHT}$ distribution was 2.40 SD (~17 cm). We next compared the heights of euploid individuals with varying percentile bins of $PGS_{HEIGHT}$ to sex-chromosome aneuploidies and observed that 45,X patients were −0.81 SD (95% CI: −1.24, −0.37; $p = 3.46 \times 10^{-4}$) shorter than the 1st percentile bin (Supplementary Table 2). On the other hand, height increases caused by an additional X- or Y-chromosome were approximately equal to the effect of a $PGS_{HEIGHT}$ in the 100th (47,XYY), 99th (47,XXX), and 88th (47, XXY) percentile bins in karyotypically normal individuals (Fig. 1a).

When we applied this same approach to BMI and LDL-C (Fig. 1b, c), we again observed that the effect size of some RGD-causing variants were larger than an extreme PGS. Relative to variant-negative individuals in the 100th percentile bin of $PGS_{BMI}$, the effect of the 16p11.2 deletion on BMI was greater by 0.38 SD (95% CI: −0.02, 0.77; $p = 0.06$), however, this difference was non-significant. The difference is even more pronounced in our observation of FH-causing variants. The presence of a pathogenic *LDLR* variant was associated with a 1.84 SD (95% CI: 1.53, 2.14; $p = 1.60 \times 10^{-28}$) and a pathogenic *APOB* variant was associated with a 0.76 SD (95% CI: 0.48, 1.04; $p = 1.59 \times 10^{-7}$) higher LDL-C than variant-negative individuals in the 100th percentile bin of $PGS_{LDL-C}$. On the other hand, the effect of an extreme PGS was equivalent to rare pathogenic variants for some traits examined in our study including *MC4R* variants ($PGS_{BMI}$ 99th percentile bin), 16p11.2 duplications ($PGS_{BMI}$ 6th percentile bin), and *PCSK9* pLOFs ($PGS_{LDL-C}$ 2nd percentile bin).

**Polygenic contribution to variable expressivity.** We quantified the contribution of PGSs to variable expressivity in patients with RGD-causing variants. As shown in Table 2, in all 11 RGDs across the three experiments, the direction of the PGS effect (β) was positive and consistent with that of our healthcare-based population (two-tailed binomial signs test: $p = 9.77 \times 10^{-4}$) (Table 1; Supplementary Fig. 4).

The PGS was significant after correction for multiple testing in five RGDs and three only met a nominal significance threshold ($p_{uncorrected} < 0.05$). All six RGDs that did not meet Bonferroni correction were underpowered (<80%) for even nominal

**Table 2 Effect size of polygenic scores in patients with rare genetic disorders**

| Trait | SNPs in PGS | RGD | Beta* (95% CI) | P-Value | P-Value (Corrected) | $R^2$ | Power |
|---|---|---|---|---|---|---|---|
| Height | 1,176,426 | Euploid | 0.46 (0.45, 0.47) | $<1 \times 10^{-300}$ | — | 0.22 | — |
| | | 47,XXX | 0.52 (0.22, 0.82) | $1.33 \times 10^{-3}$ | $5.32 \times 10^{-3}$ | 0.21 | 0.89 |
| | | 47,XXY | 0.47 (0.16, 0.77) | $3.87 \times 10^{-3}$ | $1.55 \times 10^{-2}$ | 0.15 | 0.9 |
| | | 47,XYY | 0.28 (−0.21, 0.77) | $2.56 \times 10^{-1}$ | 1 | 0.02 | 0.65 |
| | | 45,X | 0.96 (0.20, 1.70) | $1.57 \times 10^{-2}$ | $6.28 \times 10^{-2}$ | 0.26 | 0.54 |
| BMI | 1,177,440 | Variant-Negative | 0.33 (0.32, 0.34) | $<1 \times 10^{-300}$ | — | 0.11 | — |
| | | MC4R Deficiency | 0.54 (0.23, 0.85) | $9.28 \times 10^{-4}$ | $2.78 \times 10^{-3}$ | 0.17 | 0.73 |
| | | 16p11.2 Deletion | 0.37 (−0.10, 0.83) | $1.12 \times 10^{-1}$ | $3.66 \times 10^{-1}$ | 0.03 | 0.6 |
| | | 16p11.2 Duplication | 0.31 (0.11, 0.52) | $3.71 \times 10^{-3}$ | $1.11 \times 10^{-2}$ | 0.15 | 0.66 |
| LDL-C | 1,189,443 | Variant-Negative | 0.28 (0.27, 0.29) | $<1 \times 10^{-300}$ | — | 0.08 | — |
| | | LDLR FH | 0.55 (0.17, 0.93) | $7.69 \times 10^{-3}$ | $3.08 \times 10^{-2}$ | 0.04 | 0.94 |
| | | APOB FH | 0.16 (−0.12, 0.45) | $2.59 \times 10^{-1}$ | 1 | 0 | 0.76 |
| | | PCSK9 FHBL | 0.38 (−0.07, 0.69) | $1.85 \times 10^{-2}$ | $7.40 \times 10^{-2}$ | 0.28 | 0.54 |
| | | APOB FHBL | 0.30 (0.06, 0.53) | $1.39 \times 10^{-2}$ | $5.56 \times 10^{-2}$ | 0.12 | 0.45 |

RGD Rare Genetic Disorder
CI Confidence Interval
PGS Polygenic Score
FH Familial Hypercholesterolemia
FHBL Familial Hypobetalipoproteinemia
*Per 1 SD increase of the PGS

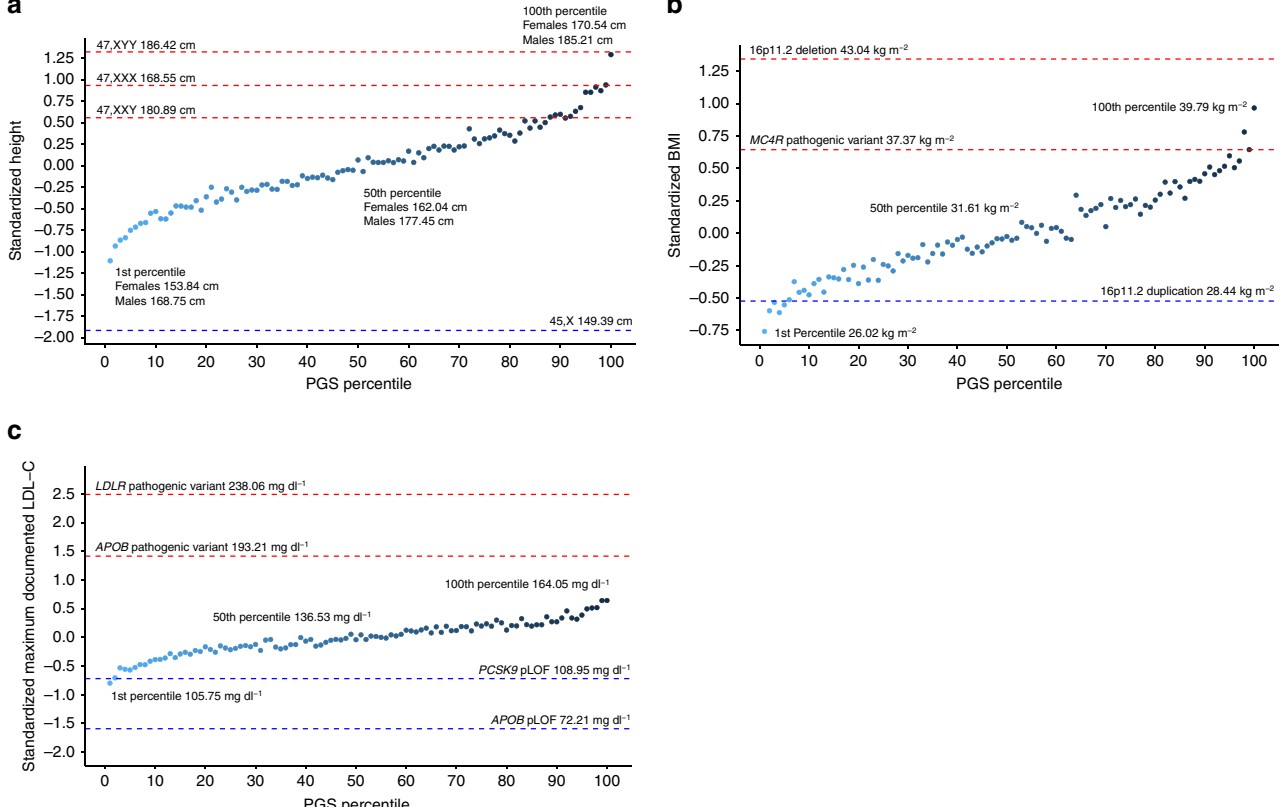

**Fig. 1** Comparison of PGS and RGDs across three quantitative traits. Mean height (**a**), BMI (**b**), and LDL-C (**c**) in variant negative individuals by percentile of the polygenic score. Points are colored from light to dark blue with an increasing PGS. Mean phenotypes of the 1st, 50th, and 100th percentile bins are indicated in non-standardized units. For comparison, the mean phenotype of RGD-causing variants are indicated as red (positive effect) and blue (negative effect) dashed lines

significance (Table 2). In our analysis, the effect size of the primary variant was not a predictor of the preservation between the PGS and the affected traits. Even in individuals with a rare and pathogenic *LDLR* variant which caused on average an increase in LDL-C of ~2.5 SD, we found evidence that the PGS still correlates with variable expressivity ($p_{corrected} = 0.03$, $r = 0.2$). On the other hand, the models of the PGS in patients with pathogenic *APOB* variants were not significant even at a nominal threshold. However, for this RGD we were slightly underpowered to detect an association (Table 2). A non-parametric analysis of

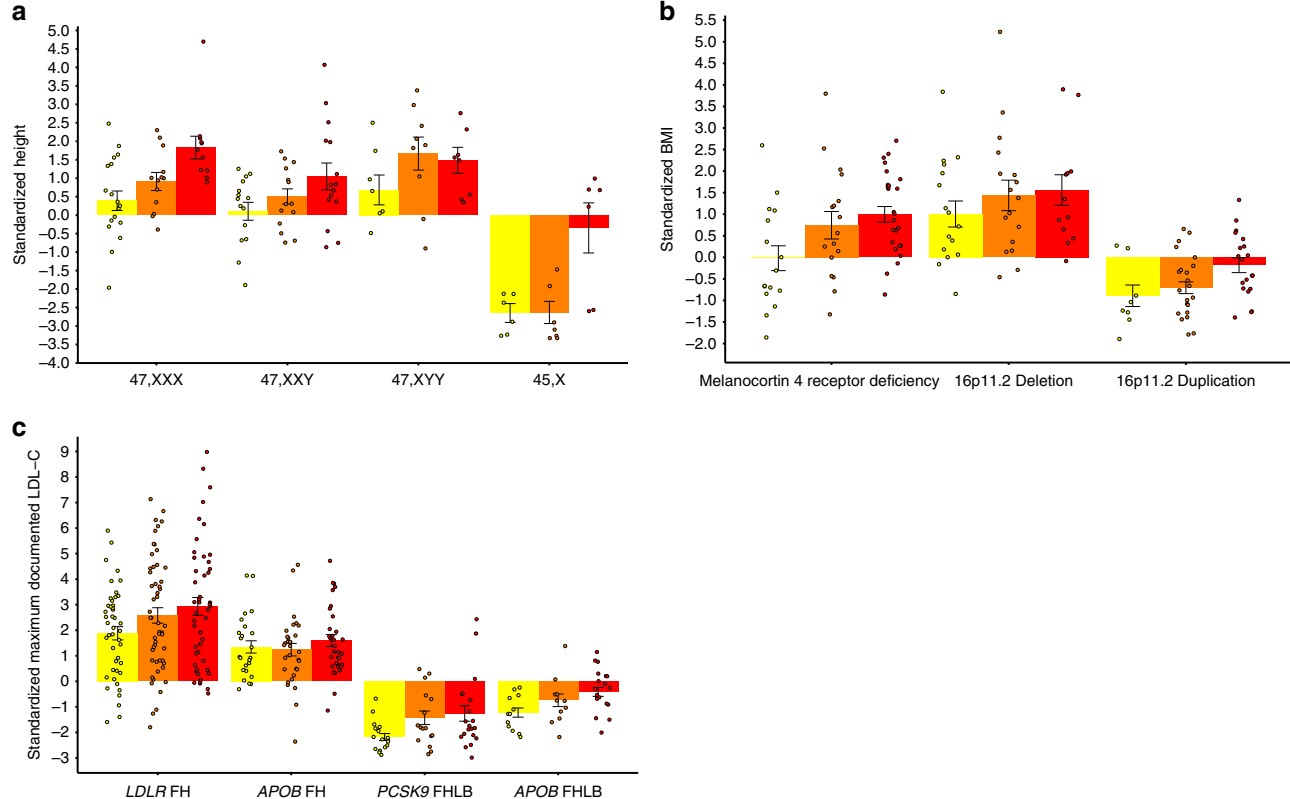

**Fig. 2** Individuals with rare genetic disorders stratified by PGS. Mean standardized height (**a**), BMI (**b**), and LDL-C (**c**) per tertile of the polygenic score within individuals with an RGD. Error bars indicate the standard error of the mean. Coloring of the bars and points yellow, orange, and red indicates a low, medium, and high polygenic score, respectively. *FH* Familial Hypercholesterolemia, *FHLB* Familial Hypobetalipoproteinemia

variable expressivity is presented in the supplement (Supplementary Table 3). Given the widespread effect of the PGS in individuals with RGDs, we recalculated effect sizes of RGD-causing variants adjusted for the respective PGS. Overall, we found that the confidence intervals generally tightened around the estimate and the resulting *p*-values were lower than the model without PGS adjustment (Supplementary Table 4).

As a descriptive measure for stratification of RGDs by clinical severity, we compared mean trait values across tertiles of PGSs (Fig. 2, Supplementary Table 5). Patients in the lowest tertile of $PGS_{HEIGHT}$ with an additional X-chromosome, 47,XXX and 47, XXY, were moderately taller (0.39 SD) or nearly the same height (0.11 SD) as euploid individuals from the general population on average (Fig. 2a), respectively. However, the relationship between variable expressivity and PGS is perhaps best exemplified in our observation that a low $PGS_{BMI}$ can effectively balance the effect of a pathogenic *MC4R* variant. The mean BMI of the 1st tertile in affected individuals is approximately equal to that of the general population (−0.02 SD) (Fig. 2b). When we examined variable expressivity of FH, we observed a 1.05 SD (45.46 mg dl^−1) difference between the first and third tertile of the $PGS_{LDL-C}$ in patients with pathogenic *LDLR* variants (Fig. 2c).

## Discussion

In this study, we used a genotype-first approach to identify patients with 11 RGDs to demonstrate their effect size in a U.S. health system-based population. The effect sizes of 45,X on height, 16p11.2 deletions on BMI, and FH-causing *LDLR* variants on LDL-C were the largest for each of the three traits studied, with 45,X and *LDLR* variants meeting statistical significance for a

more severe effect than an extreme PGS (1st or 100th percentile bins). On the other hand, we did observe in some cases an equivalence between RGD-causing variants and variant-negative individuals with an extreme PGS. Mean trait values for patients with pathogenic *MC4R* variants, *PCSK9* FHBL variants, and an additional X-chromosome (47,XXX and 47,XXY) were all equivalent to a high or extreme PGS.

Our within-gene analyses of the most penetrant causes of autosomal dominant FH (*LDLR* and *APOB*) revealed a significantly larger effect size of pathogenic variants than any percentile bin of the $PGS_{LDL-C}$. Patients with pathogenic *LDLR* variants had an LDL-C~2 SD (~75 mg dl^−1) higher than variant-negative individuals in the 100th percentile bin of the $PGS_{LDL-C}$. This stands in contrast to an analysis by the NHLBI TOPMed Lipids Working Group, who reported a ~30 mg dl^−1 effect size for both a pathogenic variant in a Mendelian hypercholesterolemia gene (*LDLR*, *APOB*, *PCSK9*, *ABCG5*, *ABCG8*, and *LDLRAP1*) and an extreme $PGS_{LDL-C}$ (>95th percentile) in European Americans[29]. While we agree that an extreme $PGS_{LDL-C}$ in variant-negative individuals can approach some clinically-relevant hypercholesterolemia variants, we did not find evidence that an extreme $PGS_{LDL-C}$ is equivalent to rare pathogenic variants in two canonical FH genes, which have specialized guidelines for screening and clinical management[30]. Across the two studies, the difference in the effect size of a rare pathogenic variant and $PGS_{LDL-C}$ may be the result of our narrow list of canonical FH genes considered or stricter criteria we applied to pathogenic/likely pathogenic (P/LP) missense variants (at least two-stars in ClinVar). Importantly, we found that including variants with assertions of pathogenicity, but lacking assertion criteria or evidence of a consensus interpretation across clinical

laboratories in ClinVar (zero- and one-star variants), can reduce the overall effect size of the presence of a pathogenic variant in a Mendelian disease gene (Supplementary Figure 5).

In another study examining the risk for developing coronary artery disease (CAD), the $PGS_{CAD}$ was proposed to identify individuals with a risk equivalent to a FH-causing variant[13]. The authors report that the odds ratio of a high $PGS_{CAD}$ (top 8%) in the UK Biobank and FH-variants in DiscovEHR were both approximately equal to a three-fold increase in risk with respect to the general population[31]. The DiscovEHR FH-variant effect size referenced in that comparison is based on an inclusive definition of FH variants that lumps known and predicted pathogenic variants in three canonical FH genes (LDLR, APOB, and PCSK9) for the risk of CAD across the lifespan. However, in the same study the presence of a pathogenic variant in LDLR was associated with a 7-fold increase of risk for premature CAD (defined as males ≤ 55 years and females ≤ 65 years) and when further limiting the analysis to LDLR loss-of-function variants, a 10-fold increase was observed[31]. This observation underscores the importance of variant classification and disease onset of Mendelian disorders, which should be taken into consideration if effect sizes of rare pathogenic variants are to be used as a benchmark for demonstrating the clinical utility of PGSs.

There are challenges in comparing the effect size of PGSs and rare pathogenic variants across cohorts when the affected phenotype is modifiable. High LDL-C levels are a risk factor for CAD and often treated with statins (57.17% have electronic healthcare record (EHR) documentation of statin usage in testing cohort), which may attenuate the effect size of the $PGS_{LDL-C}$ and rare pathogenic variants, resulting in differences between treated and untreated populations. In our analysis, the 100th percentile bin of the $PGS_{LDL-C}$ was associated with a 3.03 (95% CI: 2.30, 4.14; $p = 9.01 \times 10^{-14}$) times higher odds of having a documented statin prescription (cases = 13,597; controls = 18,147) relative to the remainder of the testing cohort, demonstrating a significant risk for hypercholesterolemia. Despite an increase in treatment for hypercholesterolemia, individuals in the 100th percentile bin of the $PGS_{LDL-C}$ had a 1.53 (95% CI: 1.13, 2.05; $p = 5.03 \times 10^{-3}$) higher odds of EHR-documented CAD (cases = 5539; controls = 16,803). This suggests that while individuals with a polygenic susceptibility to hypercholesterolemia are more likely to be treated with statins at some point, there is still a residual increase in CAD risk. This may be the result of non-compliance, under-dosing, underprescribing or limitations to statin efficacy. Here, we approximated the uncontrolled state by using maximum EHR-documented LDL-C as a phenotype, consistent with a previous study of FH in DiscovEHR[31]. However, further study into the predictive accuracy of PGSs in the context of ongoing medical-treatment is warranted[32].

Most importantly, our study demonstrated that variable expressivity can have a polygenic component for RGDs affecting three distinct quantitative traits. This finding corroborates over 50 years of research on the relationship between RGDs and familial background[1,7,9,33,34]. These past studies demonstrated that inclusion of the biparental mean with the RGD provides a more accurate prediction of clinical severity in the proband than the primary variant alone. Here, we were able to effectively substitute PGSs for the biparental mean and confirm that transmitted alleles outside the primary locus significantly contribute to this phenomenon. Notably, we observed correlations between PGSs and phenotypic severity of several RGDs previously reported to exhibit variable expressivity, including obesity caused by melanocortin 4 receptor deficiency (MC4R pathogenic variants), tall stature in 47,XXX and 47,XXY, and hypercholesterolemia caused by pathogenic LDLR variants[2,35–37]. For most of the RGDs in the present study (sex-chromosome aneuploidies, melanocortin 4

receptor deficiency, 16p11.2 CNVs, and FHBL), we provide novel examples of using a PGS to explain variable expressivity. We demonstrate that PGSs can account for the differences between asymptomatic and severely affected individuals with the same RGD-causing variant. For example, we observed that the average BMI of patients with pathogenic MC4R variants and a favorable PGS (1st tertile of the $PGS_{BMI}$) were approximately equal to the population mean −0.02 SD (Fig. 2b), while those with an unfavorable PGS (3rd tertile of the $PGS_{BMI}$) had an average BMI 1 SD above the mean.

MyCode is one of few cohorts with large-scale array-based genotype and exome-sequence data in a healthcare-based population. This resource provided data to simultaneously measure the effect sizes of PGSs and RGD-causing variants in the same study population rather than relying on published estimates. With 92,455 patients and utilizing a genotype-first approach, we were able to identify 609 unrelated individuals with RGDs. A genotype-based definition of RGDs from the general population is critical for studies of variable expressivity since clinical ascertainment (i.e., recruitment limited to specialty clinics) of probands typically narrows the study population to the most severely affected. Analyses without an accurate representation of genotypic and phenotypic diversity in their study population can therefore underestimate the role of genetic background on variable expressivity. As a case in point, the difference we observe between the first and third tertiles of the $PGS_{LDL-C}$ (47.62 mg dl^-1, Fig. 2c) in individuals affected with pathogenic LDLR variants was three times greater than the value reported from an analysis performed on FH subjects attending a lipid clinic (15.5 mg dl^-1)[15].

The analyses presented here have limitations that should be addressed in future PGS-based studies of variable expressivity. First, to acquire enough samples to achieve adequate power, we combined different classes of variants (i.e., missense, pLOF, CNV, etc.). In a post-hoc analysis, we re-tested the association between the $PGS_{LDL-C}$ only including a subset of individuals with a putative loss-of-function (pLOF) variant in LDLR ($n = 56$). As expected, LDLR pLOFs have a larger effect size, 3.22 SD (95% CI: 2.96, 3.48) SD, than when including P/LP missense variants, 2.54 SD (95% CI: 2.38, 2.70). Even in this small sample, the PGS predicted an increase in LDL-C of 0.66 SD (95% CI: 0.03, 1.29; $p = 0.04$) per unit increase. This result suggests that analyses limited to a single variant class will likely reproduce similar findings described here, although further study is warranted. Second, since patients in DiscovEHR are primarily of European descent (95.9% based on genetic ancestry) we limited our analysis to patients of this genetic ancestry[38]. The majority of individuals in GWAS are of European ancestry, and accuracy of the PGS weakens with increased genetic distance from the discovery GWAS study population, thus reducing the utility in individuals of non-European descent[39]. Therefore, larger GWAS of under-represented populations are critical to further generalize the potential clinical significance of the PGS in patients with RGDs. Third, further study into the relationship between rare pathogenic variants and common variation may reveal deviations from additivity. In an exploratory analysis, we tested for interactions between the PGS and the presence of an RGD causing variant (Supplementary Table 6). However, the results for these tests were non-significant and likely require even larger cohorts of individuals with RGDs to achieve the needed statistical power.

As discussions about incorporating PGSs into clinical care and risk prediction gain momentum, our study shows that special attention should be given to their clinical utility in patients affected with RGDs. In clinical practice, RGD-causing variants are reported to the patient based on variant interpretation standards and the known or suspected pathogenicity of the reportable variant(s). In this study, we have shown that addition of a PGS

could improve risk stratification, predict clinical severity, and help guide preventive care recommendations for individuals with pathogenic *MC4R* and *LDLR* variants. This may be especially important for the counseling of children and their parents, who, among the conditions explored here, may be advised to begin obesity prevention strategies, consider growth hormone treatment for height, or take lipid-lowering medications. Here, we were able to demonstrate that common, polygenic factors contribute significantly to variable expressivity, however, further research is necessary to specifically identify which PGSs and in what contexts meet an appropriate standard for use in clinical care[14].

In order to implement PGSs as part of the management of patients with RGDs, evidence-based guidelines, in conjunction with decision support tools in the electronic health record, could assist clinicians providing care for these patients[11,39]. Specialized communication strategies, both verbal and visual, can be developed to explain the relevance of the PGS in the context of a rare variant of larger effect size[40]. For example, the incorporation of the PGS into the return of pathogenic *LDLR* variants may help bridge the reported disconnect in understanding the difference between a diagnosis of high cholesterol and FH[40]. In the special case of reporting pathogenic variants to individuals without the phenotype of hypercholesterolemia, refining risk assessments with a PGS can provide a more precise explanation for their current disease status and a more accurate appraisal of future risk[41]. In addition, while the biparental mean has provided information regarding clinical expression in some patients with RGDs, the promise of PGSs to provide more accurate risk prediction information warrants future studies into clinical translation so that precision genomic counseling can be provided to patients and their families. This information has the potential to motivate patients with an unfavorable PGSs in addition to a pathogenic variant to adhere to medications, especially if their affected relatives may have milder phenotypes due to PGS at lower tails of risk. In conclusion, these results generalize a potential use for PGSs in medically vulnerable individuals across RGDs where appropriate early interventions can dramatically change the course of disease.

There are potential biases and patient characteristics related to the use of the MyCode cohort that should be considered in the interpretation of our results. First, patient-participants in MyCode were recruited from the Geisinger healthcare system. MyCode undersamples adults younger than age 30 years and oversamples patients in the age range of 60–89[21]. Consequently, the oversampling of older individuals may bias us toward the mild end of the clinical spectrum for some RGDs that cause a significant increase in mortality, notably 45,X and FH. Nevertheless, eligibility to participate does not depend on any particular diagnosis or insurance policy and outside these limitations the MyCode participants provide a reasonably good sampling of the Geisinger adult patient population. Secondly, our selection of RGDs for this study was limited by the sample size of individuals currently sequenced through DiscovEHR ($n = 92,455$), and therefore we could only study RGDs with a high enough prevalence to perform within-disorder analyses. Similarly, trait selection was limited to those traits captured in the EHR as part of routine healthcare. However, several RGDs in this study have broad phenotypic effects on traits that would only be measured in a minority of individuals in specialty clinics, such as the motor and behavioral deficits caused by the 16p11.2 deletion. Future studies of the polygenic contribution to these traits could be tested through genotype-based ascertainment or in epidemiological cohorts where these are routinely measured through in-person assessments or questionnaires. Our study was underpowered to detect an association for several RGDs and we acknowledge where we failed to reject the null hypothesis of no

association and cannot definitively conclude that variable expressivity in these cases does not have a polygenic basis. Statistical power will likely be an ongoing problem for studies of specific RGDs, even with access to large-scale cohorts linked to genotypic data. For example, in the UK Biobank, only 30 individuals were identified with 45,X out of 244,848 females, likely caused by an undersampling due to an ascertainment bias towards healthy individuals[42]. Lastly, we did not provide a replication analysis of our results in an independent sample. Even with ascertainment bias, the UK Biobank is an attractive cohort for future studies of many RGDs. Exome-sequencing of the 500,000 participants is currently underway, which in combination with the already generated array data, can be utilized in an analogous way to the genotype-first approach of variable expressivity described here[43].

The mechanism behind the polygenic contribution to variable expressivity of RGDs is an intriguing topic that warrants further study. At the moment, the mechanism behind PGS in the general population remains elusive, although hypotheses, such as the omnigenic model have been proposed[44]. Interaction studies between RGD-causing variants and polygenic scores may potentially reveal insights into this area of research. Here, our data, and other published data, across RGD/trait pairs is most consistent with a model of the PGSs and monogenic variants being additive and independent. However, larger sample sizes of these and other RGDs may reveal significant interaction effects.

## Methods

**Setting and study participants**. All patient-participants provided written or electronic informed consent for participation in the MyCode™ Community Health Initiative under a protocol approved by the Geisinger Institutional Review Board. This protocol allows use of participant's EHR and other clinical data and the ability to generate genomics data linked to this clinical data. The MyCode cohort is a U.S. healthcare-based population based in central and northeastern Pennsylvania and has been described in detail previously[21]. DiscovEHR is a collaboration between Geisinger and the Regeneron Genetics Center to generate whole-exome sequence (WES) and single-nucleotide polymorphism (SNP) genotype data paired with participants clinical data for discovery research[45].

**Sample inclusion**. Genomic analyses for this study were limited to unrelated individuals of European descent with EHR-documentation of the phenotypes studied (Supplementary Fig. 1). We removed one sample from each pair of first- and second-degree relationships (PI_HAT > 0.1875) identified by genome-wide identity by descent calculations using Plink v1.9[46]. Additionally, one sample from each third-degree degree related pair (PI_HAT > 0.09875) was removed if the relationship was identified in a first-degree family network reconstructed by PRIMUS[47]. We limited our analysis to non-Latino white patients of European descent confirmed by genetic ancestry and EHR-documentation of ethnicity and race. Genetic relatedness and ancestry of DiscovEHR and the methodology for assessment has been reported in detail elsewhere[38]. Four samples were excluded from this analysis based on phenotype: two samples 10 cm tall indicating a data entry error and two samples below 121 cm tall after a review of EHR data revealed the ICD-9 code for phantom limb syndrome (353.6), which may indicate height measurements were complicated by amputation of lower limbs.

**Exome sequencing and variant calling**. Sample preparation and whole exome sequencing were performed at RGC[38,45]. Exome capture was performed using either NimbleGen SeqCap EZ VCRome ($n = 61,062$) or xGen Integrated DNA Technologies ($n = 31,393$) kits according to the manufacturer's recommended protocol. The captured DNA was PCR amplified and quantified by qRT-PCR (Kapa Biosystems). Multiplexed samples were sequenced using 75 basepair (bp) paired-end sequencing on an Illumina v4 HiSeq 2500 to a coverage depth sufficient to provide greater than 20× haploid read depth of over 85% of targeted bases in 96% of samples (~80× mean haploid read depth of targeted bases). Raw sequence data from each Illumina Hiseq 2500 run were uploaded to the DNAnexus platform for sequence read alignment and variant identification. Reads were aligned to the GRCh38 reference genome using BWA-mem[48]. Variants were called separately by VCRome and xGen exome capture in groups of 200, including a pseudo-sample with all single nucleotide variants (SNVs) and indels identified across all samples. Resulting variants were filtered on QC metrics: indels were excluded if quality by depth (QD) < 5.0 and depth (DP) < 10; SNVs were excluded if QD < 3 and DP < 7; and all variants with an allelic balance below 0.15 were excluded. The final VCF

was annotated with dbNSFP (version 3.3a), Variant Effect Predictor (VEP) (version 93)[49], and ClinVar (downloaded January 26, 2018)[50]. Copy number variants (CNVs) were called from WES using the CLAMMS algorithm[51,52].

**Genotype arrays.** All patients included in this study were genotyped on either Illumina HumanOmniExpress (HOEE; n = 59,499) or Global Screening Array (GSA; n = 31,062) platforms. Upon completion of scanning, raw GTC files from each Illumina iScan run was gathered in local RGC buffer storage and uploaded to the DNAnexus platform for automated analysis. Sample-level Plink PED files, signal files (LRR and BAF) and QC metrics were generated. Following completion of cohort scanning, a project-level VCF (pVCF) and Plink BED/BIM/FAM files were generated for downstream analysis. The files were created utilizing the merge function in Plink. Data analysis and review ensured the REF and ALT alleles, for SNPs and INDELs, are reflected accurately in both the plink and PVCF files. By platform, genotypes were imputed to the Human Reference Consortium Panel[53] (r1.1) via the Michigan Imputation Server[54]. Variants passing an INFO > 0.7 on both platforms were merged and strict quality control was applied to the combined dataset. A full description of our workflow describing pre- and post- imputation quality control is presented in Supplementary Fig. 6.

**Variant curation for RGDs.** To identify individuals with FH-causing variants, we screened all samples based on variant annotations in two genes, *LDLR* (NM_000527.4) and *APOB* (NM_000384.2), using gene-level criteria. We defined all "high impact" variants called by VEP as putative loss-of-function variants (pLOFs) and CNVs that overlapped with a previously reported tandem duplication in *LDLR* (exon 13 to 17) as predicted pathogenic. Initiation codon variants (start loss) were not considered for pathogenicity. Missense variants were included if they were annotated in the ClinVar database as "Pathogenic", "Likely pathogenic", or "Pathogenic/likely pathogenic" (P/LP). Variants only affecting the last or penultimate exon were not considered for pathogenicity. To assess the utility of using ClinVar review status of P/LP variants to identify *bona-fide* FH-causing variants, we compared the LDL-C of patients with zero-star (P/LP submission(s) without assertion criteria), one-star missense (single P/LP submission with assertion criteria), and two-star missense variants (multiple P/LP submissions with assertion criteria) in *LDLR*. We observed that the LDL-C of patients with zero- and one-star missense variants was over 1.5 SDs lower than those with two-star missense variants (Supplementary Figure 5). Since this finding may be indicative of incorrect assertions or variants of weak penetrance, we limited our analysis of pathogenic missense variants to those with at least two stars in ClinVar. In the remainder of the text, predicted and known pathogenic variants are collectively referred to as pathogenic. Two *APOB* missense variants from a previous study of FH in DiscovEHR[31], p.Arg3527Gln (ClinVar ID: 17890) and p.Arg3527Trp (ClinVar ID: 40223), were also included in the present analysis as FH-causing variants. We note that while p.Arg3527Trp is currently annotated in ClinVar as "conflicting interpretations", since one research lab interpreted it as a variant of uncertain significance (VUS), several clinical genetic testing laboratories submitted "pathogenic" assertions and no laboratory provided "Benign" or "Likely Benign" assertions of interpretation.

We also considered FH-causing gain-of-function missense variants in *PCSK9* (NM_174936.3) for inclusion in the present analysis. We identified two *PCSK9* variants present in DiscovEHR, p.Arg215His (ClinVar ID: 201127) and p. Ser465Leu (ClinVar ID: 403292), called P/LP by at least one clinical genetics laboratory in ClinVar. However, we only identified a single individual with the p. Arg215His variant in DiscovEHR and the interpretation of the p.Ser465Leu variant in ClinVar is in conflict between two clinical genetics laboratories. Since we were unable to identify an adequate sample size for gain-of-function *PCSK9* variants with strong evidence for pathogenicity, this gene was not included as a monogenic form of FH in our study.

Two monogenic forms of familial hypobetalipoproteinemia (FHBL) were included in this analysis. pLOFs in *APOB* or *PCSK9* were defined as pathogenic FHBL-causing variants using the same gene-level criteria used for FH-causing variants described above.

We included two RGDs of obesity in our study, melanocortin 4 receptor deficiency caused by pathogenic *MC4R* (NM_005912.2) variants and recurrent 16p11.2 BP4-BP5 deletions (GRCh38/hg38 chr16: 29,638,675-30,188,534)[10,24]. Recurrent 16p11.2 BP4-BP5 duplications of the same region were included in our study as an RGD cause of leanness[10]. All 16p11.2 BP4-BP5 CNVs that passed this criterion overlapped at least 95% of the pathogenic region, consistent with non-homologous allelic recombination mediated by segmental duplications. 16p11.2 deletions and duplications called by CLAMMS were orthogonally confirmed by external clinical testing or by confirmation testing using the array data. 16p11.2 CNVs were called from the array data using PennCNV[55] or by manual inspection of the signal intensity data within 16p11.2 region.

Our criteria for predicted and known *MC4R* pathogenic SNVs were the same as *LDLR* described above: pLOFs or P/LP missense variants with two stars in ClinVar. A predicted pathogenic deletion of the full *MC4R* gene (GRCh38/hg38 chr18:60371350-60372776) called in the WES data by CLAMMS was also included. We excluded one pLOF (p.Leu328Ter) variant near the 3′ end of the *MC4R* transcript demonstrated to result in a normal functioning receptor[35].

To identify patients with a sex-chromosome aneuploidy, we calculated the per sample median Log R ratio (mLRR) (Supplementary Fig. 2) of X- and Y-chromosomes genotyped on the HOEE and GSA SNP array panels. Y-chromosome markers were included if they fell within the male-specific region of the Y-chromosome (GRCh37/hg19 chrY:2694521-59034049). The GSA contains 16,879 X-chromosome and 1456 Y-chromosome markers and the HOEE contains 21,112 X-chromosome and 1165 Y-chromosome markers. Following a previous study of sex-chromosome aneuploidy[56], we limited our calling of aneuploidies to high-quality genotype arrays as inferred from chromosome 1 mLRR SD and removed samples with a value above 0.28 (n = 7,250). Additionally, one HOEE genotyping batch showed exceptional variance in X-chromosome intensity and was excluded from further analysis (n = 2,666). Since the distribution of X- and Y-chromosome density differed across the GSA and HOEE platforms (Supplementary Figure 2), mLRR thresholds for aneuploidy were considered separately.

45,X and 45,X/46,XX cases were identified as EHR-documented females with extremely low X-chromosome mLRRs, which corresponded to a value below −0.28 and −0.20 on the HOEE and GSA platforms, respectively. Females with loss of X-chromosome or Y-chromosome at conception or early in development can be identified by a strong deviation chromosome-wide from expected levels of heterozygosity (0.5) relative to a 46,XX karyotype. Samples with age-related loss of X-chromosome also have decreased mLRRs. However, the BAF profile does not deviate from expected levels of heterozygosity (0.5) since a random X-chromosome is lost at each event[42]. Therefore, to identify germline mosaicism we confirmed complete loss of heterozygosity in all cases by visual inspection of BAF chromosome-wide (Supplementary Fig. 3). To estimate the proportion of 45,X-to-46,XX in 45,X/46,XX cell lines, we calculated the ratio of the sample mLRR relative to the minimum female 45,X mLRR on the corresponding array[42]. In our final set of cases, we only included samples with at least a 0.60 45,X/46,XX ratio since this value approximated the upper bound of the male 46,XY X-chromosome mLRR distribution. The mLRR thresholds used to identify 47,XXY; 47,XXX; and 47,XYY are described in Supplementary Table 7. We excluded two individuals with 48, XXXY and one individual initially characterized with a 47,XXX karyotype who after inspection of the X-chromosome logR and BAF profiles revealed an isochromosome for the long arm of the X-chromosome.

All SNVs included in this study are shown in Supplementary Data 1. The prevalence of all variants underlying RGDs in DiscovEHR before applying sample inclusion criteria are shown in Supplementary Table 8.

**Construction and tuning of PGSs.** We constructed three PGSs, a per-sample genetic load of trait-increasing alleles based on SNP effect sizes reported in publicly available GWAS summary statistics[25,26]. The PGSs were developed from these data using the Bayesian computational algorithm LDPred, which adjusts the weighting of the effect sizes with the turning parameter (ρ), a prior based on linkage disequilibrium (LD) and fraction of non-zero effects[57]. Similar to a prior analysis of PGSs in the UK Biobank, we optimized PGSs in a validation cohort for which we randomly selected 10,000 DiscovEHR individuals[13]. The ρ with the most variance explained (coefficient of determination, $R^2$) in the validation cohort was then selected for use in the testing cohort (Supplementary Fig. 7). PGSs were calculated with weights generated by LDPred using PLINK version 1.9[46].

**Phenotype definitions.** Quantitative phenotypes were developed from outpatient EHR measurements (Supplementary Fig. 8). Median values were calculated for height and BMI phenotypes, and the maximum documented value was calculated for LDL-C. Participants were assigned a CAD case or control status using an electronic phenotyping algorithm described in a previous publication[58].

In brief, CAD criteria was defined as a history of coronary revascularization in the EHR, acute coronary syndrome, ischemic heart disease, or exertional angina with angiographic evidence of obstructive coronary atherosclerosis.

**Statistical analyses.** All quantitative traits were pre-adjusted for age, six principal components of ancestry, and genotype batch separately by sex. Association testing between RGD-causing variants/PGSs with standardized quantitative measurements were performed with linear regression models and Spearman's rank correlation in the testing cohort. For the multiple testing criterion, PGS analyses in three quantitative measurements were considered as separate experiments. Here, statistical significance was defined at a Bonferroni corrected *p*-value for the number of RGDs tested within each trait (Height: 4; BMI: 3; LDL-C: 4) and nominal significance was considered at an uncorrected *p*-value of 0.05. Statistical power for linear models was calculated using the *pwr* R package using the maximally performing $R^2$ in the validation cohort. The association between cardiovascular disease and an extreme PGS$_{LDL-C}$ was with logistic regression including the same covariates as the linear models. *P*-values reported in the text are from two-tailed tests of association using linear regression models unless otherwise stated. All analyses measuring the effect sizes of RGD-causing variants and PGSs presented in the main text were performed in the testing cohort (Supplementary Fig. 1).

Interactions between the PGS and RGD-causing variants were calculated by testing for equality of the PGS beta-estimates in RGD-positive and RGD-negative

models using the normally distributed test statistic below[59]:

$$Z_{diff} = \frac{\beta_{RGD-} - \beta_{RGD+}}{\sqrt{se(\beta_{RGD-}) - se(\beta_{RGD+})}} \qquad (1)$$

All statistical analyses were conducted using R version 3.4.1.

**Reporting summary**. Further information on research design is available in the Nature Research Reporting Summary linked to this article.

## Data availability

Height and BMI polygenic scores were derived from publicly available GWAS summary statistics provided by the Genetic Investigation of ANthropometric Traits (GIANT) consortium available through their website. The LDL-C polygenic score was derived from publicly available GWAS summary statistics provided by the Global Lipids Genetics Consortium (GLGC) available through their website. Additional information on the DiscovEHR study is available through the DiscovEHR browser. Data from this study can be obtained by contacting the investigators directly.

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

## Acknowledgements

We would like to thank all patient-participants engaged in the MyCode Community Health Initiative. We would also like to acknowledge the members of the Geisinger-Regeneron DiscovEHR Collaboration who have been critical in the generation of the data used for this study (Supplementary Note 3). We thank Dr. Hunt Willard and Dr. Les Kirchner for their critical review of the manuscript; Joe Leader, Nate Person, Celeena R. Jefferson, and Dustin Hartzel for their assistance with the DiscovEHR variant and phenotypic data; and Dr. Sarah Pendergrass and Dr. Jeffrey Kidd for their thoughts on X- and Y- chromosome analyses. Funding/Support: This study was supported by grants R01MH074090 and R01MH107431 from the National Institute of Mental Health (Dr. Christa Lese Martin and Dr. David Ledbetter).

## Author contributions

Concept and design: M.T.O., D.H.L., and C.L.M. Acquisition, analysis or interpretation of data: M.T.O., M.A.K., A.C.S., D.H.L., and C.L.M. Drafting of the manuscript: M.T.O., M.A.K., A.C.S., D.H.L., and C.L.M.

## Competing interests

D.H.L. is a scientific consultant to Clear Genetics, Inc. and Natera, Inc. The remaining authors declare no competing interests.
