## [Peer Review File · Nature Communications]

Reviewers' Comments:

Reviewer #1:

Remarks to the Author:

In this article, the contribution of common genetic variation to the variable expressivity of eleven different rare genetic disorders is assessed. Studies of this nature require massive sample sizes and the authors have leveraged Geisinger's DicovEHR cohort of 92,455 patients to identify 609 individuals carrying mutations underlying rare genetic disorders. The study focuses on rare genetic disorders affecting height, BMI and cholesterol levels because a) an appreciable number of variants underpinning genetic disorders affecting these traits can be found in a cohort of this size and b) the role of common genetic variation underpinning these traits has been powerfully examined via GWAS. The main findings of the work are 1) improved estimates of the effect size of the highly-penetrant variants underlying variation in these traits and 2) a clear demonstration that the polygenic background on which these variants occur is important in shaping the phenotypic presentation of the patient.

I would like to begin by congratulating the authors on both the scale and scope of the study. I thought the manuscript was well written and an enjoyable read. The utility of PGS for risk prediction is a hot topic in the field of human genetics right now, and this manuscript makes a timely and important contribution to this.

My comments on the manuscript are:

1) There are a number of places where claims are made based on weak/no statistical evidence. For example, the effect of the 16p11.2 deletion on BMI was not significantly greater than that for the variant-negative individuals in the 100th percentile of PGS(LDL-C) ($P=0.06$, and I think this may be the uncorrected P-value too). Also, for 45, X individuals there was not statistical support for your claim that the relationship between the PGS and LDL-C levels was preserved because the p (corrected) for this test was 0.06. Please correct the interpretation of these two analyses in light of the results of your statistical tests not achieving significance. I don't believe this overly affects your general conclusions or the impact of the paper.

2) When testing for association between the quantitative traits of interest and the variants underpinning the rare genetic disorders, it would be good to understand how you went about deciding which variables to include as covariates in the analysis. For example, it looks like sex is accounted for when considering height, but not for BMI. What about smoking status when considering BMI? Finally, were there any other differences between the carriers of the mutations and the euploid individuals that need to be considered as covariates to reduce confounding?

3) At the beginning of the results section, you estimate the effect of rare pathogenic variants on the traits of interest. It would be interesting to see how much these effect size estimates change if you then correct for the relevant polygenic score.

4) On a more philosophical note, at several points in the manuscript you seem to suggest that the clinical utility of PGS is dependent on showing that, in the extremes, they have an equivalently large effect on complex traits as rare pathogenic variants. I recognise that, due to the way Khera et al framed their work, this has become a standard way of thinking about the potential utility of PGS. In my opinion, this is a rather lazy way of assessing the utility of PGS'. Using your EHR data, can you, for example, show how the incidence of CAD is correlated with high LDL-C PGS. Presumably, there are young people within your cohort who may not be aware they are at significantly increased risk of CAD and should be being closely monitored or put on statins early? It would be good to know what phenotypes are enriched within these individuals at the extremes of the PGS' relative to those around the average (and across the age range). Are any of these phenotypes for which early intervention is possible? I suppose my point is that for some of these traits the 100th percentile of the PGS is less extreme than the variant underpinning the rare

genetic disorder, but that doesn't necessarily mean that knowing one is in this 100th percentile is not clinically useful. Given your EHR data, this is something you could be uniquely placed to look into. This would be a welcome advance to the field in my opinion.

Best wishes,
Carl A. Anderson.

Reviewer #3:

Remarks to the Author:

This is an interesting paper investigating interplay between rare and common variants. They find evidence supporting that common, polygenic factors of complex traits contribute to variable expressivity of rare genetic disorders. This is an important finding, but there are some issues that should be addressed:

- 1) Why were 11 PGS selected – no rationale is given – seems a bit ad hoc. There are for example several CNVs with specific phenotype effects that can be included in the study
- 2) How were the polygenic traits selected? This also seems to be somewhat random. For example, why not investigate Waist Hip Ratio? Other blood lipids? Now there seem to be an effect of all PGS, but maybe there are some that don't work?
- 3) Power is a problem – with only 609 individuals, and six RGDs that did not meet Bonferroni correction were underpowered. Why did they include groups in the analyses that were underpowered?
- 4) PGS is reported as prediction accuracy – what is the explained variance (Nagelkerke r^2)?
- 5) They report statistical association (Effect Size of Polygenic Scores in RGD) but I couldn't find the results from interaction test – is there significant interaction between rare and common variants?
- 6) They claim that the sample is a primarily unselected clinical population, but this is difficult to understand. What are the criteria for becoming a patient in this health system? There seems to be a clear bias in sampling, related to health insurance.
- 7) The accuracy for the PGS LDL-C in the testing cohort was 8% which seems low. Why was this not higher?
- 8) What are the mechanisms involved? If you have another x-chromosome (47,XXX) this X-linked genes are often not accounted for in the GWAS, as most PGS SNPs are based on GWAS of the autosomes. Further, common variants have been shown to add to the effect of rare variants in other diseases, such as Alzheimer's disease (i.e. Desikan Plos Med 2017).
- 9) They combined missense, pLOF, CNV, while more specific analyses revealed stronger effects in sub-analysis, but then statistical power was reduced. It would be of interest to add samples from other cohorts to increase power – such as the UK Biobank sample (CNVs, X-chromosome variants). Then the findings will be more convincing.

Reviewers' comments:

We hope to receive your revised paper within three months

Reviewer #1 (Remarks to the Author):

In this article, the contribution of common genetic variation to the variable expressivity of eleven different rare genetic disorders is assessed. Studies of this nature require massive sample sizes and the authors have leveraged Geisinger's DiscovEHR cohort of 92,455 patients to identify 609 individuals carrying mutations underlying rare genetic disorders. The study focuses on rare genetic disorders affecting height, BMI and cholesterol levels because a) an appreciable number of variants underpinning genetic disorders affecting these traits can be found in a cohort of this size and b) the role of common genetic variation underpinning these traits has been powerfully examined via GWAS. The main findings of the work are 1) improved estimates of the effect size of the highly-penetrant variants underlying variation in these traits and 2) a clear demonstration that the polygenic background on which these variants occur is important in shaping the phenotypic presentation of the patient.

I would like to begin by congratulating the authors on both the scale and scope of the study. I thought the manuscript was well written and an enjoyable read. The utility of PGS for risk prediction is a hot topic in the field of human genetics right now, and this manuscript makes a timely and important contribution to this.

My comments on the manuscript are:

1) There are a number of places where claims are made based on weak/no statistical evidence. For example, the effect of the 16p11.2 deletion on BMI was not significantly greater than that for the variant-negative individuals in the 100th percentile of PGS (LDL-C) ($P=0.06$, and I think this may be the uncorrected P-value too). Also, for 45, X individuals there was not statistical support for your claim that the relationship between the PGS and LDL-C levels was preserved because the p (corrected) for this test was 0.06. Please correct the interpretation of these two analyses in light of the results of your statistical tests not achieving significance. I don't believe this overly affects your general conclusions or the impact of the paper.

We agree with the reviewer on these specific claims in our manuscript and we have taken several steps to address this in the supplement and main text. To clarify the difference in effect size between an extreme PGSs and RGD, we now include a supplemental table (Supplemental Table 2) that includes an individual test for each RGD with uncorrected and corrected p-values. In the text, we have removed claims of statistical significance of a larger effect size of 16p11.2 deletions and the 100th percentile. The association between the PGS_{HEIGHT} and height in 45,X is now referred to as nominally significant ($p < 0.05$).

2) When testing for association between the quantitative traits of interest and the variants underpinning the rare genetic disorders, it would be good to understand how you went about deciding which variables to include as covariates in the analysis. For example, it looks like sex is accounted for when considering height, but not for BMI. What about smoking status when considering BMI? Finally, were there any other differences between the carriers of the mutations and the euploid individuals that need to be considered as covariates to reduce confounding?

In general, we selected covariates representative of the relatively simple models used in the original discovery GWAS from which we derived our polygenic scores. Sex was accounted for in all analyses: all three traits were pre-adjusted for age, sex, principal components, and genotype batch. We reported descriptive statistics of height separately by sex since the values would be highly influenced by the sex ratio in our study population. To address the reviewer's concern about smoking confounding our association between the PGS and BMI, we performed a sensitivity analysis to determine if including the variable in our analysis impacted our results. Overall, we did not find an association between "ever smoking" and BMI ($p = 0.264$, $\beta = 1.86 \times 10^{-3}$) but we did detect an association with current smoking ($p = 1.28 \times 10^{-38}$, $\beta = -0.206546$) in our dataset. In variant negative individuals or individuals with RGDs (*MC4R* variants and 16p11.2 CNV), including current smoking in the model did not cause an appreciable change to the effect size or p-value of the polygenic score. All BMI RGDs maintained their significance at bonferroni correction after inclusion of the current smoking into the model.

For our height analysis, we feel that by limiting to unrelated adults of European descent and correcting for sex and ancestry we've controlled for the most likely confounders between the polygenic score and height. Height is highly heritable ($h^2 = 80\%$), so beyond genetics and the variables we've controlled for, it is unlikely that there are other variables that explain very much of the height variance in our population. Additionally, we removed samples with heights that indicate a data entry error (<10 cm tall) or a diagnosis of a limb amputation that may have interfered with taking a measurement. We don't believe there is an ascertainment bias of individuals with sex-chromosome aneuploidies that would confound our analysis. The estimated prevalence of 47,XXX, 47,XXY, and 47,XYY are between 1:500 - 1:1000 and 45,X is estimated to occur at 1:2000-1:2500, which is very close to what we have observed in DiscovEHR (Supplemental Table 9). Since these RGDs are within the accepted prevalence in DiscovEHR, this strongly suggests to us that these individuals are not limited by clinical severity or specific diagnosis.

3) At the beginning of the results section, you estimate the effect of rare pathogenic variants on the traits of interest. It would be interesting to see how much these effect size estimates change if you then correct for the relevant polygenic score.

We agree with the reviewer's suggestion, since adjusting the rare variant for the polygenic score may give a more precise estimate of the effect sizes. We performed this analysis for each RGD/Trait combination and included the results in a supplemental table (Supplemental Table 4). As expected, the confidence interval for most models tightened around the effect size and the resulting p-values were lower.

4) On a more philosophical note, at several points in the manuscript you seem to suggest that the clinical utility of PGS is dependent on showing that, in the extremes, they have an equivalently large effect on complex traits as rare pathogenic variants. I recognise that, due to the way Khera et al framed their work, this has become a standard way of thinking about the potential utility of PGS. In my opinion, this is a rather lazy way of assessing the utility of PGS'. Using your EHR data, can you, for example, show how the incidence of CAD is correlated with high LDL-C PGS. Presumably, there are young people within your cohort who may not be aware they are at significantly increased risk of CAD and should be being closely monitored or put on statins early? It would be good to know what phenotypes are enriched within these individuals at the extremes of the PGS' relative to those around the average (and across the age range). Are any of these phenotypes for which early intervention is possible? I suppose my point is that for some of these traits the 100th percentile of the PGS is less extreme than the variant underpinning the rare genetic disorder, but that doesn't necessarily mean that knowing one is in this 100th percentile is not clinically useful. Given your EHR data, this is something you could be uniquely placed to look into. This would be a welcome advance to the field in my opinion.

We agree with the reviewer that an analysis of clinical outcomes associated with the PGSs included in our study could provide further support for their clinical utility in the general population. To this end, we examined the prevalence of cardiovascular disease in 317 individuals with a PGS_{LDL-C} in the 100th percentile and compared their risk to the remainder of the population. Individuals in the 100th percentile of the PGS_{LDL-C} had a 3.03 (95% CI: 2.30, 4.14; $p = 9.01 \times 10^{-14}$) higher odds of having a documented statin prescription relative to the remainder of the testing cohort, demonstrating a high penetrance of the hypercholesterolemia phenotype. Despite an increase in treatment for hypercholesterolemia, individuals in the 100th percentile of the PGS_{LDL-C} had a 1.53 (95% CI: 1.13, 2.05; $p = 0.00503$) higher odds of having EHR-documented CAD. As shown in the two figures below, the prevalence of statin usage and risk of CAD was increased across age groups in the testing cohort at nearly all data points. This suggests that while individuals with a polygenic susceptibility to hypercholesterolemia are more likely to be treated with statins at some point, there is still a residual increase in CAD risk, which may be the result of non-compliance, underdosing/underprescribing or limitations to statin efficacy. Similarly, we observed that 317 individuals in the 100th percentile of the PGS_{BMI} have a higher odds of many treatable obesity-related conditions including type-2 diabetes 2.80 (95% CI: 2.17, 3.61; $p = 1.73 \times 10^{-15}$), hypertension 2.06 (95% CI: 1.59, 2.69; $p = 5.46 \times 10^{-8}$) and heart failure 2.29 (95% CI: 1.47, 3.43; $p = 0.0001270751$). While we are excited by these data and updated the main text, we feel that inclusion of all these results may shift the focus away from risk prediction in individuals with rare genetic disorders, which to us is the primary message of the manuscript. Our preference would be to leave some of these results to the published peer review file here.

Reviewer #3 (Remarks to the Author):

This is an interesting paper investigating interplay between rare and common variants. They find evidence supporting that common, polygenic factors of complex traits contribute to variable expressivity of rare genetic disorders. This is an important finding, but there are some issues that should be addressed:

- 1) Why were 11 PGS selected – no rationale is given – seems a bit ad hoc. There are for example several CNVs with specific phenotype effects that can be include in the study

We regret the reviewers confusion and hope to clarify our criteria for traits and rare genetic disorders selected for this study. To address the reviewers question, it should be noted that only three PGSs and 11 rare genetic disorders were selected for inclusion into our study. As stated in the text, these three traits were selected because they were 1) quantitative 2) highly heritable 3) summary statistics were available from a large genome-wide association study 4) frequently measured during routine health care and 5) affected by clinically significant variants at an appreciable frequency for within-disorder analyses. The reviewer is correct to point out that 16p11.2 deletion and duplications have effects on quantitative traits beyond BMI. Similar to other neurodevelopmental CNVs, the 16p11.2 deletion has a deleterious impact on IQ, neuromotor performance, and behavior traits. This was the focus of our previous family-based approach to variable expressivity in the Simon's VIP cohort published in JAMA Psychiatry (Moreno-De-Luca et al. 2015). However, Simon VIP probands are mainly ascertained from neurodevelopmental clinics where these measurements were captured across individuals as part of a predefined study protocol. Here, our study population was derived from a healthcare based population, where cognitive, neuromotor performance, and behaviour traits are not routinely measured. Many of individuals with 16p11.2 deletions and duplications were seen at Geisinger for healthcare unrelated to the cognitive and psychiatric effects of their primary variant. On the other hand, BMI is routinely screened and we have at least one measurement on nearly all individuals present in DiscovEHR and therefore suitable for within-disorder analyses in our healthcare-based population. To the reviewers point, we do intend to follow up our study of 16p11.2 deletions and duplications in the Simons VIP data to measure the association with the polygenic background and the variable expressivity of other affected phenotypes.

2) How was the polygenic traits selected ? This also seems to be somewhat random. For example, why not investigate Waist Hip Ratio? Other blood lipids? Now there seem to be an effect of all PGS, but maybe there are some that don't work?

We enthusiastically agree with the reviewer that this study design can be expanded beyond the traits explored here. Variable expressivity in rare genetic disorders is a widespread phenomenon and there are many potential avenues of research that polygenic scores provide. The purpose and scope of this manuscript is to generalize the finding that variable expressivity has a polygenic component and to provide insight into its clinical significance. We limited our manuscript to LDL-C, BMI, and height since they are heritable enough to provide a polygenic score that could be applied in a small sample (i.e., individuals with RGDs). These are also traits where RGDs affecting these traits are prevalent enough to perform within-disorder analyses. While waist-to-hip ratio (WHR) is sometimes measured in epidemiological studies, it is not collected as part of routine health care and we have few individuals (< 1%) in DiscovEHR with recorded WHR measurements. Other lipids could be considered as well, however, we focused on familial hypercholesterolemia given its prevalence in DiscovEHR and actionability.

Identifying RGDs with variable expressivity that fails to correlate with the PGS is certainly of interest to us and could be identified as a negative interaction as explored below. Potentially, a negative interaction with an RGD could result from large-scale disruption of a core pathway moderated by the polygenic score, which would be informative for mechanisms of both the RGD and the PGS. In this study, while we identified RGDs where we failed to reject the null hypothesis of association (e.g., *APOB*), we were unable to reject the null hypothesis of no interaction.

3) Power is a problem – with only 609 individuals, and six RGDs that did not meet Bonferroni correction were underpowered. Why did they include groups in the analyses that were underpowered?

We agree with the reviewer that low counts can present challenges to within-disorder studies of the rare genetic disorders studied here. However, our group has extensive expertise ascertaining and analyzing

rare genetic disorders. Statistical power will likely be an ongoing problem for studies of specific rare genetic disorders even in the age of large-scale cohorts linked to genotypic data. For example, in the UK Biobank, only 30 individuals were identified with 45,X out of 244,848 females, which were likely undersampled due to an ascertainment bias towards healthy individuals (Tuke et al. 2018). Despite statistical challenges, studies of these rare-genetic disorders with a population-based genotype-first approach, as presented here, provides the best vantage point for analyzing variable expressivity, since the full range of clinical severity is captured. Study populations derived from clinical cohorts are often limited to the severely affected and underestimate variable expressivity of the disorder. We acknowledge in cases where we failed to reject the null hypothesis of no association, we cannot definitively conclude that variable expressivity in these cases does not have a polygenic basis. We included statistical power in Table 2 to prevent such interpretations. Instead, we see a significant enrichment of positive associations across the 11 disorders tested including underpowered tests, which suggest that the effect is more likely to generalize beyond the disorders studied here.

4) PGS is reported as prediction accuracy – what is the explained variance (Nagelkerke r²)?

We reported R² as prediction accuracy to be consistent with the genome-wide association study that generated the statistics (Yengo et al. 2018). We have updated the text to use “variance explained” to avoid confusion.

5) They report statistical association (Effect Size of Polygenic Scores in RGD) but I couldn't find the results from interaction test – is there significant interaction between rare and common variants?

We have now included a section in the supplement investigating interactions between the PGS and rare variants. To test for interactions, we used approximately normally distributed test statistic to identify a significant difference between the PGS effect size in measured in individuals with and without RGDs:

$$Z_{\text{diff}} = \frac{\beta_{RGD+} - \beta_{RGD-}}{\sqrt{se(\beta_{RGD+}) - se(\beta_{RGD-})}}$$

Using the test above, we did not identify any PGS/RGD combinations that resulted in a significant test of association.

6) They claim that the sample is a primarily unselected clinical population, but this is difficult to understand. What are the criteria for becoming a patient in this health system? There seems to be a clear bias in sampling, related to health insurance.

We understand the reviewers concern and we apologize for not clarifying our basis for describing DiscovEHR as an unselected population. First, as a healthcare system Geisinger is a not-for-profit organization and does not have health status or insurance related restrictions for being seen for health services as a patient. Second, there is no criteria for individuals to participate in the biobanking program other than being seen as an outpatient at Geisinger. We describe DiscovEHR as an unselected healthcare-based population because eligibility to participate does not depend on a particular condition or diagnosis, participants have been enrolled from a large number of diverse clinics – including primary care, and consent rate for participation is high. We have previously shown that DiscovEHR participants provide a reasonably good sampling of the Geisinger adult patient population (Carey et al. 2016). Geisinger does offer an insurance plan (Geisinger Health Plan; GHP), however, GHP covered individuals are not preferentially recruited into DiscovEHR relative to individuals covered by other policies or uninsured. Less

than half (n=39,472) of the individuals sequenced in the 92,455 DiscovEHR samples have GHP insurance and 1,821 individuals were uninsured (1.97%). Most individuals (n=51,111) in DiscovEHR have other (non-GHP) insurance policies.

7) The accuracy for the PGS LDL-C in the testing cohort was 8% which seems low. Why was this not higher?

First, the summary statistics we used to derive our PGS_{LDL-C} were calculated from a GWAS of LDL-C, which included variants tested in ~90,000 individuals of European descent. On the other hand, the summary statistics used to derive our BMI and height polygenic scores were performed in approximately 700,000 individuals of European descent, a ~7-fold difference in sample size. For polygenic score derivation, the sample size of the GWAS influences both the power of the association test statistics and the accuracy of the estimated regression coefficients (see Chatterjee et al. 2013 for details). In our study the PGS_{LDL-C} does not capture as much of the SNP-based heritability as PGS_{Height} and PGS_{BMI}. Given the high heritability of LDL-C the predictive accuracy of the scores will improve when an LDL-C GWAS is performed on a larger cohort (e.g. ~1 million individuals).

Second, prevalent statin usage in a healthcare-based population to control hypercholesterolemia will introduce some noise into our modeling of LDL-C with polygenic scores. We inferred the untreated state by using the maximum-documented measure, which on average included over 14 years of longitudinal data. While adjusting for statins is an alternative, Geisinger is not the primary care-provider for many individuals in our study, and we may lack sensitivity for this variable in non-GHP individuals. We instead used maximum-documented LDL-C, associated with a 1.84 (95% CI: 1.80 1.90; $p < 1 \times 10^{-300}$) higher odds of at least one documented statin prescription documented in the EHR per standard deviation.

8) What are the mechanisms involved? If you have another x-chromosome (47,XXX) this X-linked genes are often not accounted for in the GWAS, as most PGS SNPs are based on GWAS of the autosomes. Further, common variants have been shown to add to the effect of rare variants in other diseases, such as Alzheimer's disease (i.e. Desikan Plos Med 2017).

Tall stature in 47,XXX is primarily driven by the triplosensitive gene, short stature *homeobox*-containing gene (*SHOX*) located in the pseudoautosomal region PAR1. Other contributions to height from X-chromosome aneuploidy outside this locus are likely much smaller in magnitude and currently not well described. Increased dosage of polygenic effects of the X-chromosome are possible but the effects cannot be measured here since the summary statistics we used for polygenic scores (Yengo et al. 2018) do not include X- or Y- chromosome markers. Instead, what amounts to essentially a *SHOX* duplication in 47,XXX (and 47,XXY) is a monogenic variant increasing height in these individuals with variable expressivity. As the reviewer points out, the height PGS presented here is the additive effect of ~1M autosomal loci of small effect. Our data across RGD/trait pairs is mostly consistent with the model of the PGSs and monogenic variants being additive and independent. The 47,XXX example presented here is consistent with that finding: monogenic loci on the X-chromosome are not expected to be different than monogenic autosomal loci and can be additive to a polygenic score generated from autosomal variants. We thank the reviewer for pointing out prior work in Alzheimer's disease and we have referenced it in the introduction of the text.

9) They combined missense, pLOF, CNV, while more specific analyses revealed stronger effects in sub-analysis, but then statistical power was reduced. It would be of interest to add samples from other cohorts

to increase power – such as the UK Biobank sample (CNVs, X-chromosome variants). Then the findings will be more convincing.

The summary statistics we used for height and BMI PGS derivation were the results of the largest GWAS (n = ~700,000 individuals) performed to date for those traits. However, most individuals in the GWAS (n=450,000) were UK Biobank participants and therefore applying our height and BMI polygenic scores would no longer be out-of-sample with respect to the discovery GWAS and statistically invalid. Outside of the UK Biobank, DiscovEHR is actually one of the few cohorts large enough to recruit enough individuals with the rare genetic disorders studied here in an unbiased ascertainment approach.

Reviewers' Comments:

Reviewer #1:

Remarks to the Author:

I liked this manuscript the first time I reviewed it and the changes made in response to my comments, and those from the other reviewer, have only improved it. It makes an important contribution to a topic of lively debate at present. I do feel the paper will influence thinking in the field. The results the authors present seem robust and the statistical analyses are appropriate.

Minor point:

1. "Even in individuals with a rare and pathogenic LDLR variant which caused on average an increase in LDL-C of ~ 2.5 SD, we found evidence that the PGS still correlates with variable expressivity ($p_{\text{corrected}} = 0.03$)." Please also provide the correlation coefficient.

Best wishes, Carl Anderson.

Reviewer #3:

Remarks to the Author:

The authors have mainly addressed the comments adequately.

However, there is a tendency to not make sufficient revisions of the manuscript text, and to rely too much on their in-house sample, despite the limitations.

The issues that were not adequately addressed were:

Rationale for study: What is possible to measure in an existing in-house sample is not a scientific rationale. The rationale must be better clarified, including the practical reasons for the selection of variables and methodology, and analytical design. The text must be revised.

Statistical Power: It is clear from the quality of the work that the group has extensive expertise in the field. Unfortunately, expertise is not of any help if the statistical power is too low. This is a clear limitation which should be better clarified in the text.

Unselected sample: It is understandable that the Geisinger research group wants to promote their Geisinger sample as unselected. The arguments for maintaining this claim are not convincing. The value of the paper will improve if they moderate the language (avoid "unselected", "unselected" population etc), and explain more details of the sample selection in the text. This will be helpful for scientists not aware of the specifics of this health care system.

Potential mechanisms involved. This is a highly interesting aspect and should be expanded in the discussion section, not only in response letter.

Replication in independent sample: the reason for not attempting replication outside of the current sample is not adequate. For example, the UK Biobank sample is freely available, and it is easy to remove overlapping individuals to generate leave-one-out results. Further, if they chose to limit the research to their in-house sample, this has to be included as a limitation in the manuscript.

Reviewer #1 (Remarks to the Author):

I liked this manuscript the first time I reviewed it and the changes made in response to my comments, and those from the other reviewer, have only improved it. It makes an important contribution to a topic of lively debate at present. I do feel the paper will influence thinking in the field. The results the authors present seem robust and the statistical analyses are appropriate.

Minor point:

1. "Even in individuals with a rare and pathogenic LDLR variant which caused on average an increase in LDL-C of ~ 2.5 SD, we found evidence that the PGS still correlates with variable expressivity (pcorrected = 0.03)." Please also provide the correlation coefficient.

Best wishes, Carl Anderson.

-We have added this data to the text.

Reviewer #3 (Remarks to the Author):

The authors have mainly addressed the comments adequately.

However, there is a tendency to not make sufficient revisions of the manuscript text, and to rely too much on their in-house sample, despite the limitations.

The issues that were not adequately addressed were:

Rationale for study: What is possible to measure in an existing in-house sample is not a scientific rationale. The rationale must be better clarified, including the practical reasons for the selection of variables and methodology, and analytical design. The text must be revised.

-We revised the text in the introduction in an effort to clarify rationale behind our study design.

Statistical Power: It is clear from the quality of the work that the group has extensive expertise in the field. Unfortunately, expertise is not of any help if the statistical power is too low. This is a clear limitation which should be better clarified in the text.

Unselected sample: It is understandable that the Geisinger research group wants to promote their Geisinger sample as unselected. The arguments for maintaining this claim are not convincing. The value of the paper will improve if they moderate the language (avoid "unselected", "unselected" population etc), and explain more details of the sample selection in the text. This will be helpful for scientists not aware of the specifics of this health care system.

Potential mechanisms involved. This is a highly interesting aspect and should be expanded in the discussion section, not only in response letter.

Replication in independent sample: the reason for not attempting replication outside of the current sample is not adequate. For example, the UK Biobank sample is freely available, and it is easy to remove overlapping individuals to generate leave-one-out results. Further, if they chose to limit the research to their in-house sample, this has to be included as a limitation in the manuscript.

-We have added a substantial limitations paragraph in the discussion that details potential biases of DiscovEHR. We have also included a discussion on potential mechanisms. The text has been copied here for the reviewers convenience:

The analyses presented here were restricted to the MyCode cohort and potential biases and limitations should be considered. First, patient-participants in MyCode were recruited from the Geisinger healthcare system. Recruitment into MyCode undersamples adults younger than age 30 years and oversamples patients in the age range of 60–89²¹. Consequently, the oversampling of older individuals may bias us toward the mild end of the clinical spectrum for some RGDs that cause a significant increase in mortality, notably 45,X and FH. Nevertheless, eligibility to participate does not depend on any particular diagnosis or insurance policy and outside these limitations the MyCode participants provide a reasonably good sampling of the Geisinger adult patient population. Secondly, our selection of RGDs for this study was limited by the sample size of individuals currently sequenced through DiscovEHR (n=92,455), and therefore we could only study RGDs with a high enough prevalence to perform within-disorder analyses. Similarly, trait selection was limited to those traits captured in the EHR as part of routine healthcare. However, several RGDs in this study have broad phenotypic effects on traits that would only be measured in a minority of individuals in specialty clinics, such as the motor and behavioral deficits caused by the 16p11.2 deletion. Future studies of the polygenic contribution to these traits could be tested through genotype-based ascertainment or in epidemiological cohorts where these are routinely measured through in-person assessments or questionnaires. Our study was underpowered to detect an association for several RGDs and we acknowledge where we failed to reject the null hypothesis of no association and cannot definitively conclude that variable expressivity in these cases does not have a polygenic basis. Statistical power will likely be an ongoing problem for studies of specific RGDs, even with access to large-scale cohorts linked to genotypic data. For example, in the UK Biobank, only 30 individuals were identified with 45,X out of 244,848 females, likely caused by an undersampling due to an ascertainment bias towards healthy individuals⁴². Lastly, we did not provide a replication analysis of our results in an independent sample. The UK Biobank is an attractive cohort for future studies of RGDs. Exome-sequencing of the 500,000 participants is currently underway, which in combination with the already generated array data, can be utilized in an analogous way to the genotype-first approach of variable expressivity described here⁴³.

The mechanism behind the polygenic contribution to variable expressivity of RGDs is an intriguing topic that warrants further study. At the moment, the mechanism behind PGS in the general population remains elusive, although hypotheses, such as the omnigenic model have been proposed⁴⁴. Interaction studies between RGD-causing variants and polygenic scores may potentially reveal insights into this area of research. Here, our data, and other published data, across RGD/trait pairs is mostly consistent with a model of the PGSs and monogenic variants being additive and independent. However, larger sample sizes of these and other RGDs may reveal significant interaction effects.